# Experimental realisation of multi-qubit gates using electron paramagnetic resonance

Edmund J. Little[1], Jacob Mrozek[2], Ciarán J. Rogers [1], Junjie Liu [2], Eric J. L. McInnes [1], Alice M. Bowen [1] ✉, Arzhang Ardavan [2] ✉ & Richard E. P. Winpenny [1] ✉

Quantum information processing promises to revolutionise computing; quantum algorithms have been discovered that address common tasks significantly more efficiently than their classical counterparts. For a physical system to be a viable quantum computer it must be possible to initialise its quantum state, to realise a set of universal quantum logic gates, including at least one multi-qubit gate, and to make measurements of qubit states. Molecular Electron Spin Qubits (MESQs) have been proposed to fulfil these criteria, as their bottom-up synthesis should facilitate tuning properties as desired and the reproducible production of multi-MESQ structures. Here we explore how to perform a two-qubit entangling gate on a multi-MESQ system, and how to readout the state via quantum state tomography. We propose methods of accomplishing both procedures using multifrequency pulse Electron Paramagnetic Resonance (EPR) and apply them to a model MESQ structure consisting of two nitroxide spin centres. Our results confirm the methodological principles and shed light on the experimental hurdles which must be overcome to realise a demonstration of controlled entanglement on this system.

In a classical computer information is stored as 'bits', the state of which is either 0 or 1, which are manipulated by Boolean logic gates. The quantum analogue of a classical bit is a 'qubit', which can exist not only as 0 s or 1 s but as superpositions of the two states. For any system to be considered a viable candidate for quantum computing it must be possible to realise a set of universal quantum logic gates, from which any quantum logic operation can be constructed (or at least simulated)[1]. This requires the implementation of at least one multi-qubit gate, the simplest of which are two-qubit operations such as CNOT, CPHASE, and √SWAP gates[2]. Quantum state tomography measures the expectation values of a set of operators which form a complete basis of the Hilbert space, and could also detect the quantum states of qubits. Appropriate pulse design for pulsed EPR spectroscopy is sufficient for tomography. All useful two-qubit gates can be implemented by a combination of an 'entangling gate', by which a separable state is converted to a maximally entangled state, and single-qubit gates. Thus, the demonstration of single-qubit gates, an entangling gate, and subsequent quantum state tomography is sufficient to demonstrate a universal set of gates[2].

The intrinsic spins of nuclei and electrons makes them natural candidates for the physical realisation of qubits, and quantum logic operations can be realised by manipulation of their spin states by nuclear magnetic resonance (NMR) and electron paramagnetic resonance (EPR), respectively[3–5]. The implementation of an entangling gate between nuclear spins by NMR was first demonstrated in liquid solutions of chloroform and cytosine ensembles, and in the case of chloroform the density matrix of the coupled $^1H$ and $^{13}C$ nuclear spins was determined by quantum state tomography. However, the large identity component of the density matrix, arising from working in the high temperature limit, meant that only "pseudo-entangled" states were accessible[6,7]. Pseudo-entanglement was also demonstrated in an electron-nuclear ensemble system generated by x-ray irradiation of a

[1]Photon Science Institute and School of Chemistry, The University of Manchester, Oxford Road, M13 9PL Manchester, UK. [2]Clarendon Laboratory, University of Oxford, Parks Road, OX1 3PU Oxford, UK. ✉e-mail: alice.bowen@manchester.ac.uk; arzhang.ardavan@physics.ox.ac.uk; richard.winpenny@manchester.ac.uk

malonic acid crystal[8], and within the spin states in $^{15}$N@C$_{60}$[9]. By working at low temperatures and introducing a polarisation transfer step between the electron and nuclear spins, it was possible to initiate a sufficiently pure starting state that real electron-nuclear entanglement was confirmed in an ensemble of phosphorus donors in silicon[10,11]. Interest in identifying systems in which all qubits are electron spins is motivated by the observation that, owing to their larger Zeeman couplings compared to nuclear spins, a sufficiently pure starting state may be obtained in thermal equilibrium at readily achievable temperatures.

Many physical systems have been proposed as electron spin qubits including nitrogen-vacancy centres in diamonds and quantum dots in silicon, though the top-down synthesis of these spin centres makes precise control of inter-spin coupling difficult, complicating their use as multi-qubit structures[12]. This led to the proposal of Molecular Electron Spin Qubits (MESQs), in which bottom-up synthesis allows for more precise control of inter-spin interactions[4]. Progress towards using MESQs as single qubits has been remarkable, particularly in performance of the Grover algorithm in a single magnetic molecule[13], and in the extension of coherence times from the microsecond to the millisecond regimes[14]. The problem of scaling remains a challenge for all electron spin qubits, whether fabricated by top-down or a bottom-up approach.

The simplest possible multi-qubit MESQ is a molecular dimer in which two electron spins are coupled by their dipolar interactions, however for the qubits to be manipulated independently it is ideal to be able to 'switch' the inter-spin interaction off[15]. While many candidates have been proposed, both with and without switchable interactions, no multi-electron-qubit MESQ gates have yet been demonstrated experimentally as far as we are aware[16,17]. (We note the recent demonstration of quantum teleportation in an electron spin system, mediated by an optically excited entangled radical pair[18]. However, here there is no control over how the entanglement is generated, it only happens by virtue of the selection rule for the optical excitation of the radical pair.)

In this article we present the transfer of NMR quantum computing methods to EPR, using a bis-nitroxide model system (Compound **1**, Fig. 1, synthesis and characterisation are given in Supplementary Note 1) to demonstrate the pseudo-entanglement and subsequent tomography of two weakly coupled electron spins. The fidelity of the entangling gate and state determination are critically assessed, through which we determine the primary causes of error accumulation and discuss these in terms of the feasibility of using MESQs as multi-qubit gates. Our protocol is designed around the hierarchy of energy scales usually encountered in such chemically engineered two-molecular-spin-qubit molecules: $\frac{\hbar}{T_m} > J > \frac{\hbar}{T_{manip}}$ where $T_m$ is the single-qubit phase coherence time, $J$ is the inter-qubit interaction energy, and $T_{manip.}$ is the single-qubit operation time.

**Fig. 1 | Skeletal structure of compound 1.**

## Generation of entanglement

The maximally entangled states of a system of two coupled spin-1/2 particles include the Bell states $|\phi^{\pm}\rangle = \frac{1}{\sqrt{2}}(|00\rangle \pm |11\rangle)$ and $|\psi^{\pm}\rangle = \frac{1}{\sqrt{2}}(|01\rangle \pm |10\rangle)$, the density matrices of which are given in the Cartesian basis by $|\phi^{\pm}\rangle\langle\phi^{\pm}| = \frac{1}{2}[\frac{1}{2}\hat{1} \pm 2\hat{S}_x\hat{I}_x \mp 2\hat{S}_y\hat{I}_y + 2\hat{S}_z\hat{I}_z]$ and $|\psi^{\pm}\rangle\langle\psi^{\pm}| = \frac{1}{2}[\frac{1}{2}\hat{1} \pm 2\hat{S}_x\hat{I}_x \mp 2\hat{S}_y\hat{I}_y - 2\hat{S}_z\hat{I}_z]$, where S and I represent the two spins. If the coupling between the two-spin centres can be approximated as a purely secular interaction, then the static Hamiltonian of the system in an external magnetic field is $\hat{H}_0 = \omega_S\hat{S}_z + \omega_I\hat{I}_z + \frac{\omega_{SI}}{2}2\hat{S}_z\hat{I}_z$[19]. Under the assumption of weak coupling ($\omega_{SI} \ll |\omega_S - \omega_I| \ll \omega_S, \omega_I$), the thermal equilibrium state of this system is given by:

$$\hat{\rho}_{eq} = \frac{1}{2}\hat{1} + \Delta p\left[\hat{S}_z + \hat{I}_z\right] + \Delta p^2 2\hat{S}_z\hat{I}_z \qquad (1)$$

where $\Delta p = \tanh(\frac{\hbar(\omega_S + \omega_I)}{4k_BT})$, $\hbar$ is the reduced Planck constant, $k_B$ is the Boltzmann constant, and $T$ is the temperature. The initial state of conventional NMR and EPR experiments is the thermal equilibrium state, which is generally highly mixed ($\Delta p \ll 1$) but can be considered to be pseudo-pure, $\hat{\rho}_{eq} = \frac{1}{2}(1 - \Delta p)\hat{1} + \Delta p\hat{\rho}^{pure}$ where $\hat{\rho}^{pure}$ is the pure state[20]. 400 MHz and room temperature (typical NMR conditions) generates $\Delta p \approx 0.00003$, whereas 9.5 GHz and 50 K (typical EPR conditions) generates $\Delta p \approx 0.005$. Measurements performed on a single-spin system with a pseudo-pure initial state are functionally identical to those performed with a truly pure initial state, with the only difference being a reduction in signal intensity. The same is not true for a two-spin system, where the expectation value of $2\hat{S}_z\hat{I}_z$ in $\hat{\rho}^{pure}$ depends on $\Delta p$. For clarity the identity element will be omitted from further discussion, as it is invariant under unitary transformations and is not measurable. As such all subsequently defined density matrices represent traceless deviation density matrices, as is common in magnetic resonance[21]. We note, however, that for a two-MESQ system at achievable temperatures and magnetic fields, the amplitude of $\hat{\rho}^{pure}$ can exceed the threshold required to demonstrate real entanglement rather than pseudo-entanglement[22]. Resonance techniques are able to detect pseudo-entanglement due to their sensitivity to small deviations from the equilibrium density matrix.

In principle entanglement between two coupled spins could be generated by driving the $|00\rangle \leftrightarrow |11\rangle$ and $|10\rangle \leftrightarrow |01\rangle$ conditional transitions with resonant pulses. This will be most convenient in the intermediate coupling regime ($\omega_{SI} \approx |\omega_S - \omega_I|$) in which case all possible transitions are spectrally addressable and have similar EPR transition matrix elements[23]. In weakly coupled systems for which the coupling is too small to yield spectrally addressable conditional transitions, it is more convenient to generate entanglement by allowing the system to freely evolve under its coupling Hamiltonian. This method is easier to implement as it is only necessary to pulse at two frequencies (resonant with S and I) and was first demonstrated in NMR using a dual-frequency Hahn-echo sequence (Supplementary Fig 1, right)[6,7].

In a single-frequency Hahn-echo (SFHE) sequence (Supplementary Fig 1, left), coherences are generated on spin S at time $0^+$, and are completely refocussed into an echo at time $2\tau$. The magnetisation of the I spin is unaffected by the pulses and as such the density matrix when the echo is formed (assuming pulses are applied along $+x$) is given by:

$$\hat{\rho}_{2\tau} = \Delta p\left[\hat{S}_y + \hat{I}_z\right] + \Delta p^2 2\hat{S}_y\hat{I}_z \qquad (2)$$

In the absence of decoherence the magnetisation is not dependent on the inter-pulse delay, $\tau$. In a DFHE experiment coherences and echoes are generated on both spins, and the application of 180° pulses at time $\tau$ prevents complete refocussing of the inter-spin coupling

**Table 1 | Spin operators which lead to single-quantum coherences, detectable as transverse magnetisation (top) under all pairs of operations**

| Operation pair | $\hat{S}_x$ | $\hat{S}_y$ | $2\hat{S}_x\hat{I}_z$ | $2\hat{S}_y\hat{I}_z$ | $\hat{I}_x$ | $\hat{I}_y$ | $2\hat{S}_z\hat{I}_x$ | $2\hat{S}_z\hat{I}_y$ |
|---|---|---|---|---|---|---|---|---|
| $1_S1_I$ | $\hat{S}_x$ | $\hat{S}_y$ | $2\hat{S}_x\hat{I}_z$ | $2\hat{S}_y\hat{I}_z$ | $\hat{I}_x$ | $\hat{I}_y$ | $2\hat{S}_z\hat{I}_x$ | $2\hat{S}_z\hat{I}_y$ |
| $1_SX_I$ | $\hat{S}_x$ | $\hat{S}_y$ | $2\hat{S}_x\hat{I}_y$ | $2\hat{S}_y\hat{I}_y$ | $\hat{I}_x$ | $-\hat{I}_z$ | $2\hat{S}_z\hat{I}_x$ | $-2\hat{S}_z\hat{I}_z$ |
| $1_SY_I$ | $\hat{S}_x$ | $\hat{S}_y$ | $-2\hat{S}_x\hat{I}_x$ | $-2\hat{S}_y\hat{I}_x$ | $\hat{I}_z$ | $\hat{I}_y$ | $2\hat{S}_z\hat{I}_z$ | $2\hat{S}_z\hat{I}_y$ |
| $X_S1_I$ | $\hat{S}_x$ | $-\hat{S}_z$ | $2\hat{S}_x\hat{I}_z$ | $-2\hat{S}_z\hat{I}_z$ | $\hat{I}_x$ | $\hat{I}_y$ | $2\hat{S}_y\hat{I}_x$ | $2\hat{S}_y\hat{I}_y$ |
| $X_SX_I$ | $\hat{S}_x$ | $-\hat{S}_z$ | $2\hat{S}_x\hat{I}_y$ | $-2\hat{S}_z\hat{I}_y$ | $\hat{I}_x$ | $-\hat{I}_z$ | $2\hat{S}_y\hat{I}_x$ | $-2\hat{S}_y\hat{I}_z$ |
| $X_SY_I$ | $\hat{S}_x$ | $-\hat{S}_z$ | $-2\hat{S}_x\hat{I}_x$ | $2\hat{S}_z\hat{I}_x$ | $\hat{I}_z$ | $\hat{I}_y$ | $2\hat{S}_y\hat{I}_z$ | $2\hat{S}_y\hat{I}_y$ |
| $Y_S1_I$ | $\hat{S}_z$ | $\hat{S}_y$ | $2\hat{S}_z\hat{I}_z$ | $2\hat{S}_y\hat{I}_z$ | $\hat{I}_x$ | $\hat{I}_y$ | $-2\hat{S}_x\hat{I}_x$ | $-2\hat{S}_x\hat{I}_y$ |
| $Y_SX_I$ | $\hat{S}_z$ | $\hat{S}_y$ | $2\hat{S}_z\hat{I}_y$ | $2\hat{S}_y\hat{I}_y$ | $\hat{I}_x$ | $-\hat{I}_z$ | $-2\hat{S}_x\hat{I}_x$ | $2\hat{S}_x\hat{I}_z$ |
| $Y_SY_I$ | $\hat{S}_z$ | $\hat{S}_y$ | $-2\hat{S}_z\hat{I}_x$ | $-2\hat{S}_y\hat{I}_x$ | $\hat{I}_z$ | $\hat{I}_y$ | $-2\hat{S}_x\hat{I}_z$ | $-2\hat{S}_x\hat{I}_y$ |

The subscripts on the operations in column 1 denote the spins on which the operation is applied.

term. As a result, the echoes are modulated by the inter-pulse delay with frequency $\omega_{SI}$:

$$\hat{\rho}_{2\tau} = \Delta p\left[\cos(\omega_{SI}\tau)\left[\hat{S}_y + \hat{I}_y\right] - \sin(\omega_{SI}\tau)\left[2\hat{S}_x\hat{I}_z + 2\hat{S}_z\hat{I}_x\right]\right] + \Delta p^2 2\hat{S}_y\hat{I}_y \tag{3}$$

If the initial starting state is pure ($\Delta p = 1$) then the density matrix after a dual-frequency Hahn echo with the inter-pulse delay $\tau = \frac{\pi}{2\omega_{SI}}$ will be:

$$\hat{\rho}_{2\tau} = -2\hat{S}_x\hat{I}_z - 2\hat{S}_z\hat{I}_x + 2\hat{S}_y\hat{I}_y \tag{4}$$

This can then be converted to one of the maximally entangled Bell states by single-qubit rotation operations (for example the $|\psi^-\rangle\langle\psi^-|$ density matrix can be generated by applying 90° pulses along $+y$ to spin S and $-y$ to spin I), and so a DFHE with an appropriately selected inter-pulse delay represents an entangling gate. However, if the initial state is not pure the result of the DFHE experiment may not be a truly entangled state, but a pseudo-entangled one:

$$\hat{\rho}_{2\tau} = -\Delta p\left[2\hat{S}_x\hat{I}_z + 2\hat{S}_z\hat{I}_x\right] + \Delta p^2 2\hat{S}_y\hat{I}_y \tag{5}$$

For sufficiently small $\Delta p$, the expectation value of $2\hat{S}_y\hat{I}_y$ is effectively zero and only the anti-phase coherences can be observed.

## Quantum state tomography

To determine the density matrix it is necessary to measure the expectation values of a set of operators which form a complete basis of the Hilbert space, which for magnetic resonance experiments is most conveniently represented in the Cartesian basis. Only single-quantum coherences can be directly detected in magnetic resonance echo experiments, and so the remaining operators must be converted to a detectable form in order to measure their expectation values. A general protocol for performing quantum state tomography on an $n$-qubit system by NMR consists of a series of experiments in which one of three operations is applied to each spin prior to detection of the free induction decay (FID) signal[24]. The choices of operations are: i. do nothing, or apply a 90° 'filter' pulse along ii. $+x$ or iii. $+y$, which we will denote 1, X, and Y, respectively. In a two-qubit system there are nine possible pairs of operations, and any Cartesian operator can be converted into a detectable single-quantum in- or anti-phase coherence by at least one combination (Table 1). While it is possible to measure all operators with just four combinations it is common practice to perform all nine experiments to allow for repeated observations.

This protocol relies on the ability to measure both in- and anti-phase coherences. However, the ensemble broadening of EPR spectra precludes the direct detection of anti-phase coherences in the weak inter-spin coupling regime that we consider here[23]. As a result, to transfer this procedure from NMR to EPR it is necessary to devise a detection sequence capable of measuring the expectation values of anti-phase coherences. To accomplish this, we draw inspiration from the 3P- and 4P-Double Electron-Electron Resonance (DEER) experiments (Fig. 2) in which the application of a pump pulse at time $t$ prevents refocussing of precession under the inter-spin coupling Hamiltonian $\frac{\omega_{SI}}{2}2\hat{S}_z\hat{I}_z$[25,26]. The presence of in-phase coherences, those which would be detectable as transverse magnetisation, before the 'pump-refocus' block will manifest as cosine oscillations in the intensity of the refocussed echo and anti-phase coherences will manifest as sine oscillations, with the oscillation amplitudes proportional to their expectation values (see the product operator calculations in and their relation to expectation values in Supplementary Note 4). This has been observed previously in 4P-DEER at low temperature and high frequency, where the initial Hahn-echo acted on an almost pure initial thermal state and so generates both in- and anti-phase coherences (Eq. 2)[27].

By appending a 'pump-refocus' block to an arbitrary pulse sequence it is possible to simultaneously measure both in- and anti-phase coherences, and by repeating the experiment with the pump and detection frequencies switched the expectation values of all eight single-quantum coherences can be observed (Supplementary Table 1). This can be further extended to perform quantum state tomography by the addition of the pairs of operations outlined above, resulting in the proposed DEER Assisted Tomography (DEERATom) procedure (Fig. 3). Note that by moving the pump pulse through the refocussing period the in- and anti-phase coherences become effectively 'phase-labelled', a technique earlier used to facilitate the quantum state tomography of a hybrid electron-nuclear system[10].

It is important to note that only spin centres on resonance with the refocussing pulse *and* whose partner spins are inverted by the pump pulse will contribute an oscillatory component to the trace, with the rest either not contributing to the recorded echo or adding an unmodulated offset to its intensity. As a result, the DEERATom procedure determines the state of a sub-population of spins, or a sub-density matrix, rather than the entire ensemble.

## Results

### Measurement of spin properties

To assess the suitability of compound **1** for (pseudo-)entanglement and subsequent tomography it is necessary to characterise the attainable one- and two-qubit gate times and compare these to its coherence time. The independent manipulation of two spins requires them to have different resonant frequencies, and so it is crucial to determine that single-qubit operations can be achieved at a range of microwave frequencies. This can be accomplished by recording the

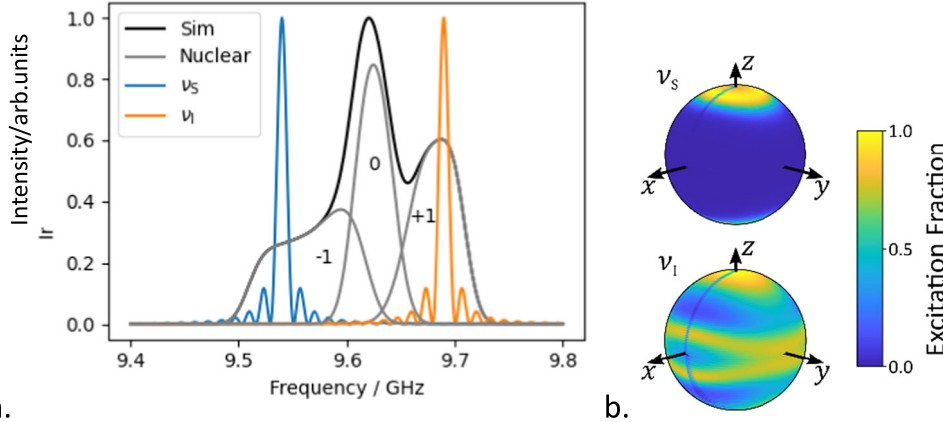

**Fig. 2 | The EPR spectroscopy of compound 1. a** Simulated frequency domain EPR spectrum of compound **1** at 342.9 mT (black) and corresponding sub-spectra corresponding to the nuclear spin states (grey), overlayed with simulated excitation profiles of 80 ns 180° pulses at $\nu_S$ (9.54 GHz, blue) and $\nu_I$ (9.69 GHz, orange).

**b** Orientations of the nitroxide spin excited by pulses at 80 ns 180° pulses at $\nu_S$ (top) and $\nu_I$ (bottom). Simulation parameters and echo-detected field-swept spectra are included in Supplementary Fig 3. All simulations were performed using the MATLAB toolbox Easyspin[44]. Definition of $\nu_I$ and $\nu_S$ as Fig. 2.

attainable nutation frequency of pulses, $\nu_1$, as a function of their carrier frequency, resulting in a resonator bandwidth (Supplementary Fig. 2). As such we determine that 32 ns one-qubit gate times can be achieved between 9.500 and 9.725 GHz, with the shortest gate time of 22 ns possible at 9.625 GHz.

For our purposes the two-qubit gate time is defined by the interaction between the two-spin centres, which we determined using the 3P-DEER experiment (Fig. 3). At X-band frequencies (≈9.5 GHz) the EPR spectra of nitroxide spins is significantly broader than the excitation bandwidth of achievable rectangular pulses. This broadening is dominated by the hyperfine interaction of the electron spin-1 $^{14}$N nucleus, causing three distinct resonances corresponding to the nuclear spin states (Fig. 2a). We choose to maximise the offset between the two frequencies, while remaining on resonance with the nitroxide spectrum, to minimise distortions which can be caused by pulse overlap in the frequency domain[28]. As such the lower frequency, $\nu_S$, corresponds to pulsing transitions within the high field $m_{^{14}N} = -1$ manifold, and the higher frequency, $\nu_I$, to the low field $m_{^{14}N} = +1$ manifold.

If the pulses used were capable of exciting all electron spins within these nuclear spin manifolds, then one third of the spin centres would be detected as the three nuclear spin states are approximately equally populated. Of these detected spins, the pump pulse would invert one third of their coupled partners, and so one ninth of the molecules in the sample would contribute an oscillatory component to the recorded trace. The pulses only excite a fraction of the spins in each manifold, and so the inter-spin interactions recorded represent an even smaller proportion of molecules corresponding to specific orientations of the nitroxide spin centres with respect to the external magnetic field (Fig. 2b)[29].

The background corrected results of the 3P-DEER experiment detected at $\nu_S$ and $\nu_I$ are almost identical (Fig. 4, raw data in Supplementary Fig. 4), and from these it can be determined that the two-qubit gate time, i.e. the first minimum in the time trace, is 250 ns. Despite being in frozen solution, compound **1** exhibits a narrow distribution of coupling frequencies centred at 2 MHz (Fig. 4, right). This is likely a result of the rigidity of the alkyne and phenyl groups connecting the two nitroxide spins, which will limit the distribution of conformations adopted in solution. The observed coupling distribution is likely further narrowed by the pulses used only exciting certain orientations of the nitroxide spin centre, and so not all orientations of the inter-spin vector will contribute to the oscillations (Fig. 4b). While the distribution of inter-spin interactions could be reduced further by orienting the sample, we find that the coupling frequency is well enough defined

in frozen solution to demonstrate a proof-of-principle entangling gate[30,31].

The coherence times were measured by 2P-Electron Spin Echo Envelope Modulation (2P-ESEEM) (Supplementary Fig. 1) and the fitted value of 6.89 μs is significantly longer than the two-qubit gate time, which is itself significantly longer than the one-qubit gate time. This relationship between operation times make compound **1** a suitable candidate for the generation and detection of entanglement via free evolution under the inter-qubit coupling.

## Implementation of an entangling gate

The generation of entanglement by a dual-frequency Hahn-echo sequence can be implemented in NMR using standard methods, both in hetero- and homo-nuclear systems[6,7]. In the hetero-nuclear case standard rectangular pulses can be used to excite each nucleus separately, as their Larmor frequencies will be significantly different[6]. The use of multiple simultaneous pulses at different frequencies is a key part of many common NMR experiments, such as the closely related Insensitive Nuclei Enhanced by Polarization Transfer (INEPT) technique[32]. In homo-spin systems the two spins can be excited selectively, either by long low-powered pulses or using shaped pulses[7].

The use of simultaneous pulses is generally avoided in EPR as overlap of pulses in time tends to lead to distortions in data. For example, the 3P-DEER sequence (Fig. 3) has a 'dead-time' at short $t$, where the initial detection and pump pulses overlap in time[26,33]. The 4P-DEER sequence (Fig. 3) circumvents the dead-time and so has almost completely superseded 3P-DEER. The principal cause of pulse distortion is the non-linear amplitude response of microwave components involved in pulse generation, which tend to saturate at a maximum outputted power[34]. To characterise the amplitude response of our spectrometer, the attainable nutation frequency of pulses was recorded as a function of their amplitude prior to amplification. In our apparatus, multiple components in the pulse generation chain exhibit nonlinearities; rather than characterising each individual component, the whole chain was calibrated at once by measuring the effect on the nutation frequency of the spin system directly[34].

We controlled the pulse amplitude using an Arbitrary Waveform Generator (AWG) over its whole range (parameterised as 0 to 100%). Our experimental set-up shows a linear relationship between amplitude and nutation frequency up to a pulse amplitude of 30% (Supplementary Fig. 5). The linear regime may extend above 30%, but a precise determination of the upper limit was not possible between 30% and 60% due to strong nuclear ESEEM oscillations caused by hyperfine coupling to $^1$H nuclei in the protonated solvent matrix.

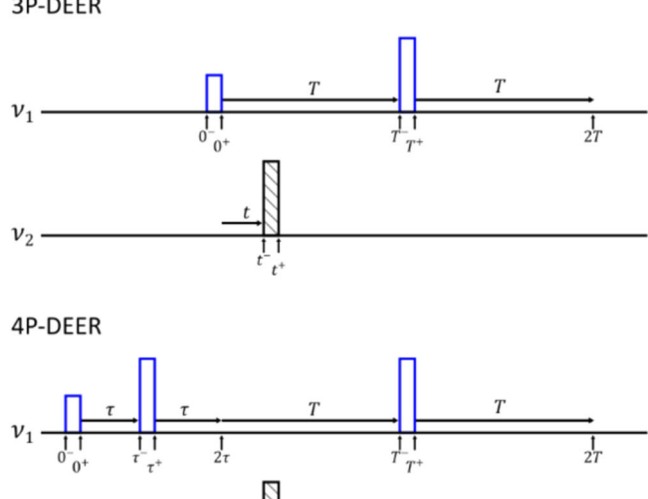

**3P-DEER**

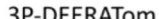

**4P-DEER**

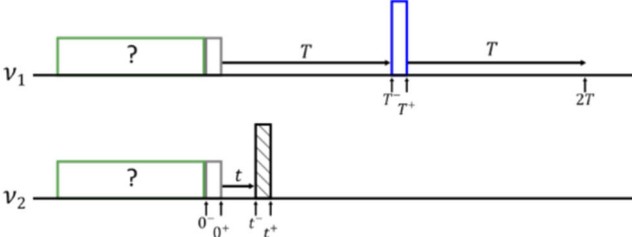

**3P-DEERATom**

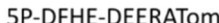

**5P-DFHE-DEERATom**

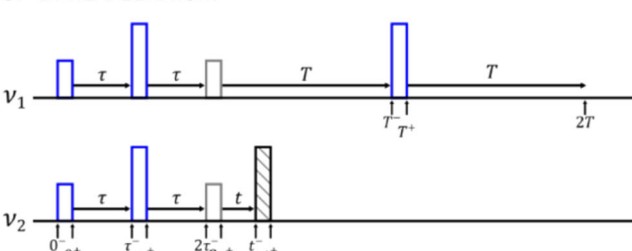

**Fig. 3 | Comparison of DEER-like pulse sequences, with coherent pulses in blue, incoherent pulses in black, and variable pulses in grey.** Arbitrary pulse sequences are in green. Pulse amplitudes and thus turning angles are represented by their heights, with 90° pulses half the height of 180° pulses. Relevant delays are indicated by horizontal arrows, and specific time points are labelled with vertical arrows for ease of comparison. The labels of the frequencies are arbitrary and so $\nu_1$ can be either $\nu_S$ or $\nu_I$, in which case $\nu_2$ would be $\nu_I$ or $\nu_S$, respectively.

By ensuring that the total amplitude of all pulses fired simultaneously stays within the linear region, distortions to pulse shapes should be minimised. This must be balanced against the desire for short pulse lengths, which both increase signal intensity and minimise evolution under the influence of $\hat{H}_0$ during pulses. We decided to use 80 ns 180° pulses, which were attainable with 20% amplitude at both $\nu_S$ and $\nu_I$. The total amplitude of both pulses when fired simultaneously is then 40%, which is very close to, if not within, the linear response regime.

While it is not possible to detect the anti-phase coherences generated by the DFHE directly, it is still possible to get a qualitative assessment of the entangling gate by recording the $\tau$ dependence of

the in-phase coherences. By incrementing the inter-pulse delay $\tau$, the intensity of the echo generated by the DFHE sequence should oscillate with frequency $\omega_{SI}$. This experiment is analogous to the 2P-ESEEM experiment (Supplementary Fig. 1), and so in addition to the desired inter-electron oscillation the recorded echo intensity will decay due to decoherence and may also be modulated by electron-nuclear couplings. These 'background' effects can be removed by division of a dual-frequency 2P-ESEEM experiment by a comparable single-frequency trace, and the result matches well with the inter-electron coupling oscillations recorded by DEER (Fig. 5). While the recorded oscillations are not direct evidence of the presence of anti-phase coherences, and thus pseudo-entanglement, they do imply that the entangling pulse sequence is working as expected. It follows that with an inter-pulse delay of 125 ns, corresponding to a quarter period of the dipolar evolution, the system will be maximally entangled.

## DEER assisted tomography

To quantitatively assess the generation of anti-phase coherences by the DFHE entangling gate with $\tau = \frac{\omega_{SI}}{4}$ (125 ns), and thus the gate fidelity, it is necessary to fully determine the state of the system. We achieve this by applying the DEERATom procedure, implemented using the 5P-DFHE-DEERATom sequence (Fig. 3). By recording the real and imaginary components of the nine pairs of operations at both frequencies we obtain thirty-six traces, each of which will contain cosine and sine oscillations with amplitudes proportional to the expectation values of the Cartesian product operators present at $2\tau^-$ (See Supplementary Note 2).

A typical DEER experiment will employ a phase cycle to account for undesired coherence pathways and imperfect spectrometer calibration. However, the requirement to measure both real and imaginary components of the traces precludes the application of a phase cycle involving the detection sequence. By using an incoherent microwave source to generate the pump pulses, it is possible to remove $t$-dependent 'echo-crossing' artefacts experimentally, but this does not affect static offsets or partially filtered magnetisation and so these must be considered in data analysis[35]. To account for static offsets we recognise that the background decay parameter, $k$, for a given sample is dependent only on the fraction of spins in the sample excited by the pump pulse[36]. All DEER and DEERATom traces recorded with the same pump pulse should have the same background decay parameter, allowing the offsets to be isolated. Since it is not possible to conveniently calibrate the excitation fraction of the filter pulses, we instead analysed the data at a range of filter pulse flip probabilities, $P_{flip}$, treating the recorded traces as the sum over filtered, partially filtered, and unfiltered magnetisation.

If the entangling gate and filter pulses function ideally then there should only be oscillations in eight traces, all of which should be out-of-phase (Supplementary Table 4). However, it is clear by visual inspection that the majority of the experimental time traces are not completely flat and thus contain some form of oscillation, and that the amplitude of these oscillations is significantly lower than in the reference DEER data (DEER date in Fig. 5 and DEERAtom data in Supplementary Figs. 7, 8 and 9). The state of the system was determined by globally fitting the data for a range of values of $P_{flip}$, with the lowest cost solution occurring at $P_{flip} = 0.42$. The pseudo-fidelity of the entanglement and tomography combined can be calculated by comparing the deviation density matrices, which we accomplish by treating the expectation values of the non-identity Cartesian product operators as vectors in the generalised Bloch sphere. In this representation the similarity between two states can be thought of as the angle between them, and we use the cosine of this angle to define the pseudo-fidelity:

$$\widetilde{\mathcal{F}}(\hat{\rho}_1, \hat{\rho}_2) = \frac{d-1}{2d}\cos\angle(\hat{\rho}_1, \hat{\rho}_2) + \frac{d+1}{2d} \quad (6)$$

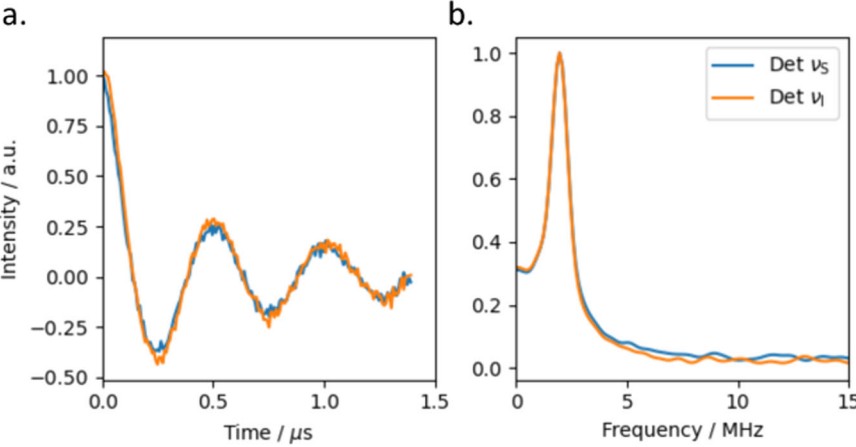

**Fig. 4 | 3P-DEER traces for 1. a** Background corrected 3P-DEER traces (left). **b** Corresponding Fourier transforms. DEER traces recorded detecting at $\nu_S$ (9.54 GHz) and pumping at $\nu_I$ (9.69 GHz) (blue) and detecting at $\nu_I$ and pumping at $\nu_S$ (orange). Definition of $\nu_I$ and $\nu_S$ as Fig. 2.

where $d$ is the Hilbert dimension, and:

$$\cos \angle (\hat{\rho}_1, \hat{\rho}_2) = \vec{\rho}_1 \cdot \vec{\rho}_2$$

$$= \sum_i^n \alpha_i \beta_i$$

$$= \mathrm{tr}(\hat{\rho}_1 \hat{\rho}_2) - \frac{1}{d}$$

where $\vec{\rho}$ is the vector representation of $\hat{\rho}$, and $\alpha$ and $\beta$ are the expectation values of the basis operators in $\hat{\rho}_1$ and $\hat{\rho}_2$, respectively (i.e. the elements of $\vec{\rho}_1$ and $\vec{\rho}_2$)[37]. Thus two directly opposing quantum states, with equal and opposite observables, will have a pseudo-fidelity of 0.25, whereas two identical quantum states will have a pseudo-fidelity of 1.00. By this metric, the pseudo-fidelity of the fitted state (Fig. 6a, b, green) to the desired state (black) is 0.788. Note that the fitted experimental state was normalised by the magnitude of the calculated expectation values because the experimental starting state is unknown. This means that the measurement is insensitive to an overall loss in magnetisation, and therefore this measurement of this pseudo-fidelity is a normalised pseudo-fidelity and an upper-bound (see further discussion later in the Relation to experiment section).

Factors leading the loss of fidelity are examined further in the Discussion section, however while discussing the fitted expectation values it is pertinent to note that due to the relatively low value found for the filter pulse flip probability, the majority of the echo intensity in all traces arises from 'leakage' of unfiltered magnetisation. If the actual filter pulse efficiency is greater than the value used in the fit, then the expectation values of observables which are not single-quantum coherences will be artificially inflated as these operators cannot be detected without being affected by a filter pulse. This will be especially problematic for experiments with filter pulses at both frequencies, in which less than 20% of the fitted signal arises from the desired coherence pathway. For example, there is no pathway by which $2\hat{S}_x\hat{I}_x$ or $2\hat{S}_z\hat{I}_z$ magnetisation can be generated by the DFHE entangling gate, and yet these are the largest expectation values in the fitted density matrix (Fig. 6a). This distortion of the fitted quantum state would lead to errors in the readout and thus a loss of fidelity.

It is possible to qualitatively assess the fidelity of the entangling gate from the results of the $1_S 1_I$ experiments (Fig. 6c, d), in which the in- and anti-phase coherences expected from the DFHE entangling gate should manifest in the time traces without any additional complications arising from filter pulses. If the entangling gate functions ideally

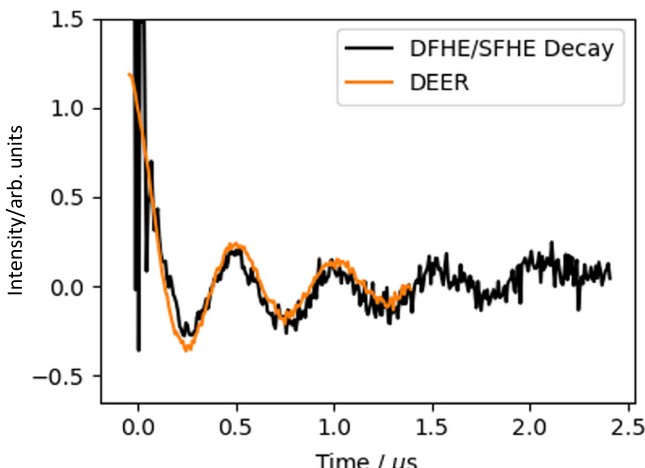

**Fig. 5 | Oscillatory part of the dual-frequency Hahn-echo decay (black), isolated by division of the raw data by the equivalent single-frequency Hahn-echo decay and manually background corrected.** This shows very similar oscillations to the comparable DEER data (orange). Time corresponds to $\tau$ for the Hahn-echo decay and $t$ for the DEER experiments. All pulses were 80 ns in length, and all data were detected at $\nu_I$ (9.69 GHz) at 343 mT, with pulses at the non-detection frequency at $\nu_S$ (9.54 GHz) where applicable. Definition of $\nu_I$ and $\nu_S$ as Fig. 2.

the real parts of these traces should be dominated by sine oscillations arising from anti-phase magnetisation. However, the traces are visibly dominated by cosine oscillations with defined minima at 0.25 μs and maxima at 0.50 μs, indicating that the recorded expectation values of the in-phase coherences are significantly larger than the desired anti-phase coherences. While it is possible that the absence of visible sine oscillations is caused by the pump-refocus sequence being more sensitive to in-phase coherences than to anti-phase coherences, we see no reason that this should be the case and so it is likely that the DFHE entangling gate is generating significantly more in- than anti-phase coherences.

## Discussion

The experimental results demonstrate a significant lack of fidelity; here we examine what we believe to be the most likely sources of loss of fidelity, considering the state generation and detection separately before simulating the experiment as a whole. The product operator calculations (Supplementary Note 4) were performed assuming ideal

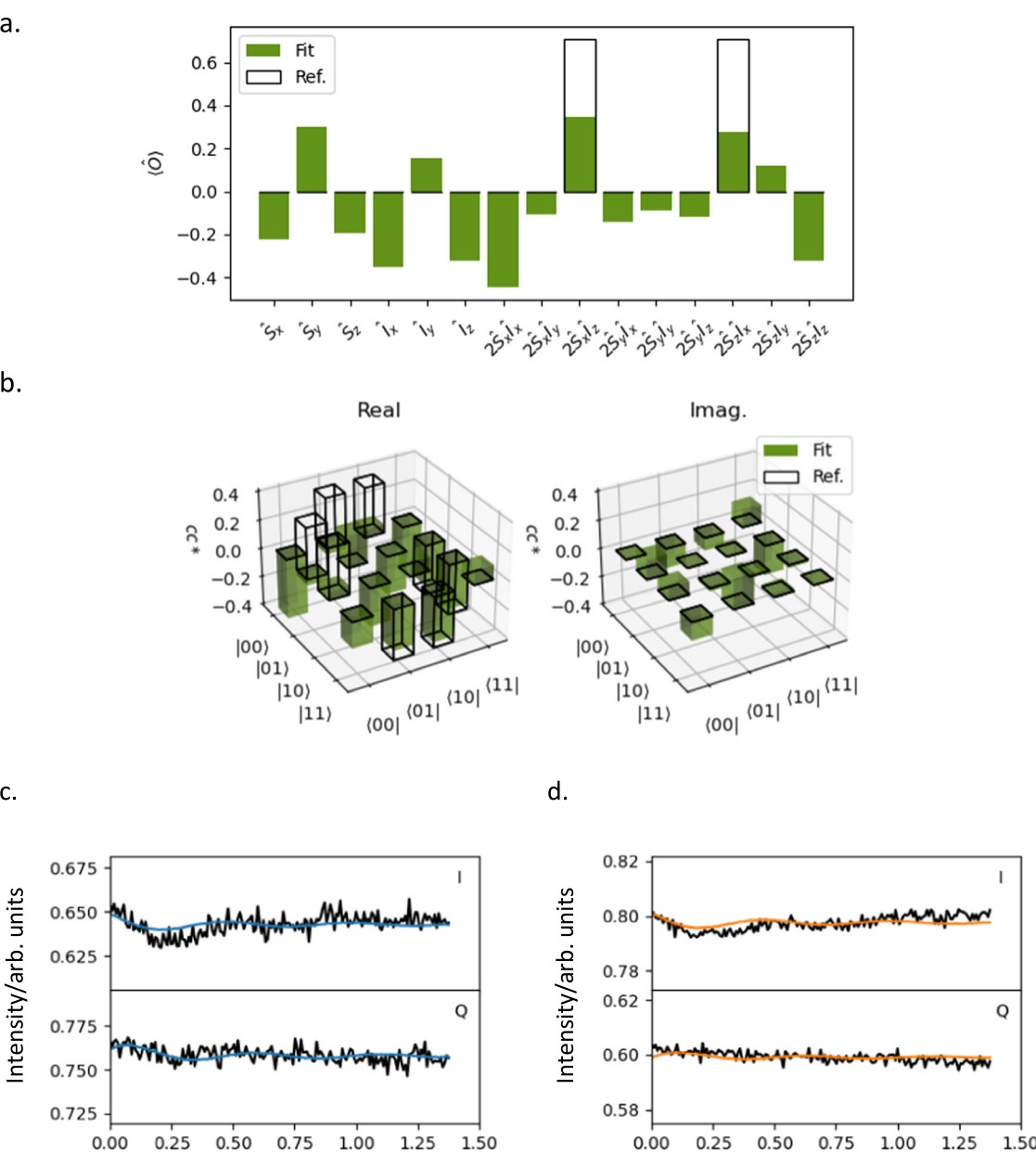

**Fig. 6 | The pseudo-fidelity of the fitted state to the desired state. a** The fitted, normalised expectation values of the Cartesian operators (green) and reference values from an ideal DFHE (black). **b** The real (left) and imaginary (right) components of the elements of the density matrix, with fitted values in green and reference values in black. **c** and **d** Experimental DEERATom time traces (black) and corresponding fits (blue and orange) recorded with the pair of operations $1_S 1_I$, detected at $\nu_S$ and $\nu_I$, respectively. Data were background corrected and normalised, with the real component denoted as I and the imaginary part as Q. Definition of $\nu_I$ and $\nu_S$ as Fig. 2.

pulses, during which precession about the static Hamiltonian is neglected. A more general approach is to consider that the static Hamiltonian is always acting on the spin system, effectively tilting the axis of nutation during pulses out of the $xy$-plane[23]. It is not convenient to model this effect within the product operator formalism as the static and driving Hamiltonians do not commute, and so we turn to numerical simulations to describe the evolution of the quantum state. Further details of the simulations are included in the Supplementary Note 5.

## State generation

**Precession during pulses.** By considering a system in which the two spins are exactly on resonance with the applied pulses, equivalent to a single molecule, the Zeeman terms in the static Hamiltonian go to zero and only the inter-spin coupling term is active. In this case the fidelity of the simulated entangling gate oscillates with frequency $\omega_{SI}$ as the inter-pulse delay is incremented (Fig. 7a). The maxima in fidelity occur at $\tau \approx \frac{(4n+1)\pi}{4\omega_{SI}}$, and there is no observable decrease in fidelity as $n$ is increased. By contrast, both pulse length and inter-spin coupling have a significant effect on the simulated fidelity, with shorter pulses and lower inter-spin coupling frequencies enhancing the fidelity of the gate (Fig. 7b). Precession under the influence of the inter-spin coupling Hamiltonian during pulses leads to the generation of $2\hat{S}_x\hat{I}_y$ and $2\hat{S}_y\hat{I}_x$ coherences, with a simultaneous increase in residual $\hat{S}_z$ and $\hat{I}_z$ magnetisation, negatively impacting the fidelity of the entangling gate. The simulation corresponding most closely to our experimental set-up (with 80 ns pulses and an inter-spin coupling of 2 MHz) demonstrated a peak fidelity of 0.996. As such the action of

a.

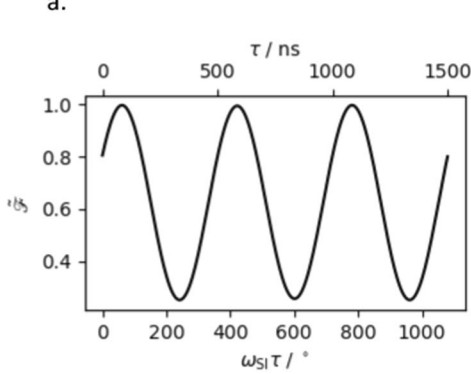

b.

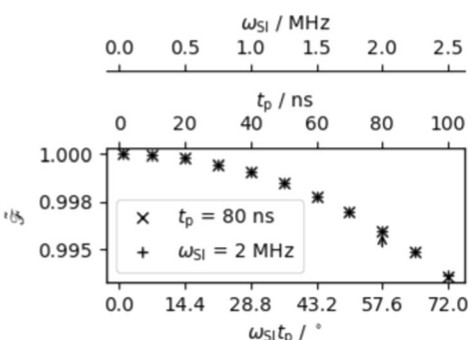

**Fig. 7 | Simulated pseudo-fidelity of the DFHE entangling gate. a** Shown as a function of nutation angle during inter-pulse delays with $t_p = 80$ ns and $\omega_{SI} = 2$ MHz. **b** Shown as a function of nutation angle during pulses, numerically evaluated by altering the inter-spin coupling frequency at either a constant pulse length (crosses) or by altering the pulse length at constant inter-spin coupling frequency (pluses). The point corresponding to the experimental set-up is marked with an arrow.

the static Hamiltonian during pulses alone does not account for the experimental loss in fidelity.

**Coupling frequency distribution.** In frozen-solution samples each molecule will have a different alignment with respect to the external magnetic field and a unique molecular conformation, leading to a distribution of coupling frequencies across the ensemble. As a result, the coherences generated by the DFHE entangling gate with a specific value of $\tau$ will represent the average over a distribution of coupling frequencies. The simplest model of inter-spin coupling is that of anisotropic interaction, corresponding to physical systems in which the inter-spin coupling is dominated by super-exchange interactions, or equivalently those in which only specific orientations of the molecule with respect to the external magnetic field are sampled. The coupling distribution was modelled as a normal distribution centred at $\mu_{\omega_{SI}} = 2$ MHz, corresponding to the maximum of the inter-spin coupling in compound **1**, with standard deviations $\sigma_{\omega_{SI}}$. As expected, the first maximum in the simulated fidelity decreases as the distribution of inter-spin couplings is broadened (Supplementary Fig 10). Through the DEER experiment, we can estimate the distribution of couplings exhibited by the sub-population probed by our experiments; the standard deviation of the inter-spin coupling was determined to be approximately 0.244 MHz (Supplementary Fig 11, Supplementary Table 2), corresponding to a fidelity of 0.989.

For the coupling between two spins to be detectable by pulse dipolar spectroscopy, the inter-spin distance must be greater than around 1.5 nm[38]. In this distance regime isotropic exchange coupling tends to be negligible, and the inter-spin interaction is generally dominated by the anisotropic dipolar interaction[39]. Assuming all non-secular terms are negligible the strength of the dipolar coupling is given by $\omega_{dd} = \omega_\perp (3\cos^2\theta - 1)$, where $\theta$ is the angle between the inter-spin vector and the external magnetic field and $\omega_\perp$ is the coupling frequency when $\theta = 90°$. While each orientation of the molecule is equally likely to be present in the sample, leading to a Pake pattern of inter-spin coupling frequencies, the orientations do not contribute equally to the recorded data due to the inherent orientation selectivity of microwave pulses (Fig. 3b). As such we can generate two models of dipolar coupling: one with uniform weights representing the sample, and one in which each orientation is weighted by the overlap of the nitroxide spectrum with the microwave pulses. The simulated fidelities of the entangling gate with $\omega_\perp = 2$ MHz for the uniform and weighted models are 0.749 and 0.821, respectively, significantly lower than the corresponding simulations considering anisotropic interaction. This decrease in fidelity is caused by interference between systems with different coupling frequencies, which

diminishes the expectation values of the anti-phase coherences. The uniformly weighted simulation is more impacted by this interference as its frequency distribution is a complete Pake pattern, whereas the frequency distribution of the spectrally weighted simulation corresponds to a partial Pake pattern.

In addition to the orientational disorder, molecules in frozen solution will adopt different conformations with different inter-spin distances. As a result, there will be a distribution in $\omega_\perp$ which will further broaden the inter-spin coupling distribution. Here we take $\omega_\perp$ to be normally distributed with mean $\mu_{\omega_\perp}$ and standard deviation $\sigma_{\omega_\perp}$. As expected, as $\sigma_{\omega_\perp}$ is increased the fidelity of both models decreases (Supplementary Fig 10). By comparing these models to our experimental data, we determine that for our sample $\sigma_{\omega_\perp}$ is 0.150 MHz and 0.133 MHz for the uniformly weighted and orientational selective models, respectively. These correspond to fidelities of 0.746 and 0.820, respectively.

**Spectral linewidth.** The variety of molecular orientations and conformations in frozen-solution samples also leads to distributions in resonant frequencies of the spin centres, with typical disordered EPR spectra being significantly broader than the bandwidth of available rectangular microwave pulses. As a result, the Zeeman terms in the static Hamiltonian cannot be neglected, leading to partial excitation by microwave pulses and dephasing of the magnetisation during delays. To assess the impact of spectral linewidth on the fidelity of the entangling gate, we model the EPR spectra as normal distributions centred at the pulse frequencies and with standard deviation $\Gamma$. As expected, the fidelity of the entangling gate decreases as the spectral linewidth increases, rapidly falling before plateauing at around 0.63 (Supplementary Fig 12). This decrease in fidelity is primarily caused by partial excitation, in particular in cases where one spin is very close to resonance, but its partner spin is much further away. In this situation the on-resonance spin packet effectively experiences a SFHE experiment, and so at the time of the echo its state almost completely consists of undesired in-phase coherences. It is not possible to define the linewidth of the nitroxide spectrum within this model, however the spectrum clearly extends over 200 MHz at X-band frequencies (Fig. 2), and so we expect that our experimental fidelity corresponds to the infinite-bandwidth limit.

**State detection.** For high-fidelity determination of any arbitrary quantum state, the amplitude of the oscillations in each DEERATom trace should be proportional to the expectation values of the relevant operators listed in Supplementary Table S4. This requires quantitative conversion of the desired magnetisation into in- and anti-phase

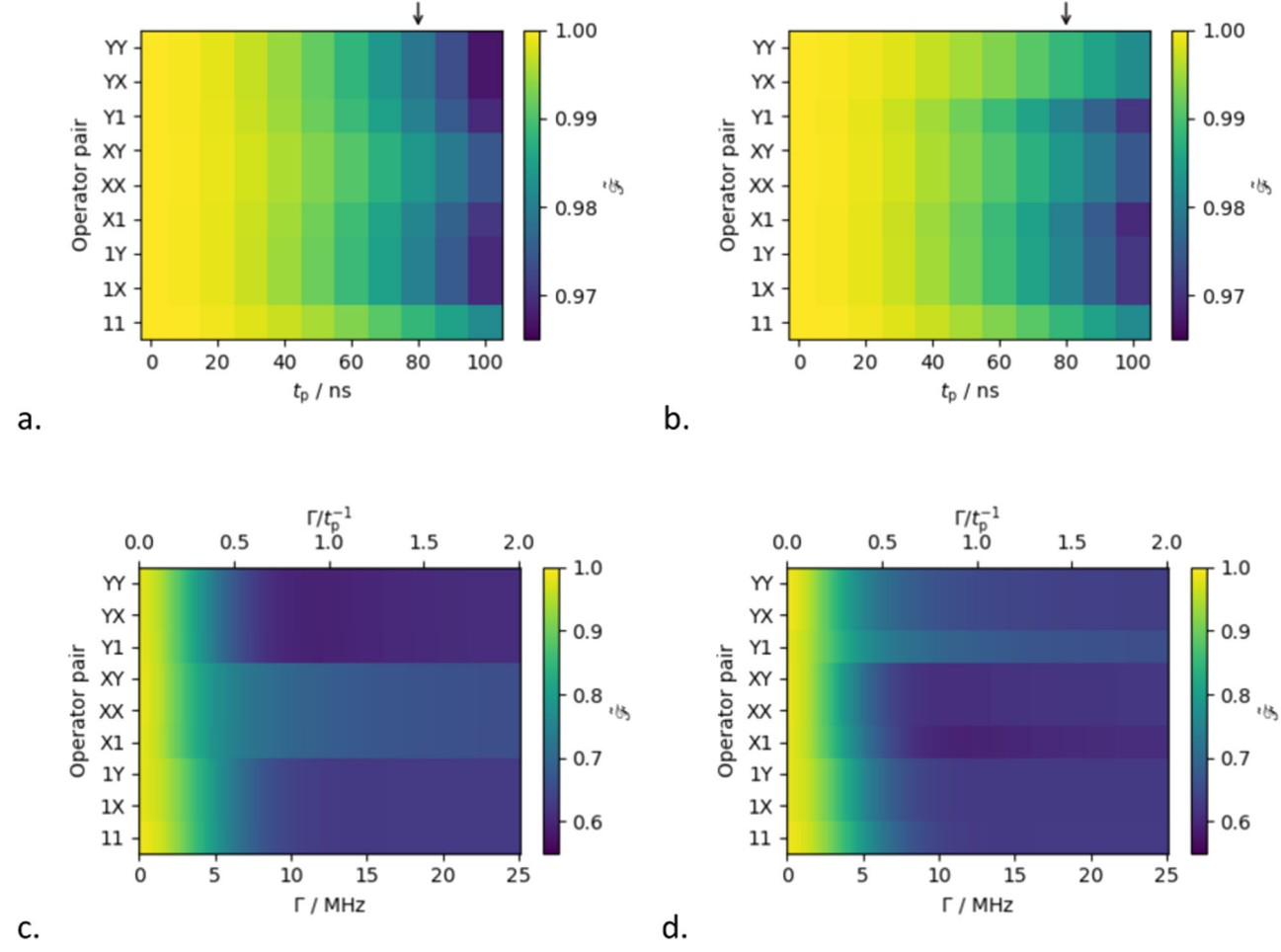

**Fig. 8 | Simulated fidelity of the filter pulses. a** As a function of pulse lengths, $t_p$, corresponding to $\hat{S}_y$ coherences. **b** as a function of pulse lengths, $t_p$, corresponding to $2\hat{S}_x\hat{I}_z$ coherences. **c** As a function of spectral linewidth, $\Gamma$, corresponding to $\hat{S}_y$ coherences. **d** As a function of spectral linewidth, $\Gamma$, corresponding to $2\hat{S}_x\hat{I}_z$ coherences. The arrows in a. and b. correspond to the pulse length used experimentally.

coherences by the filter pulses and proportional detection of these coherences by the pump-detect block.

**Filter pulses.** While a fidelity can be determined for each pair of operations acting on any arbitrary quantum state, here we limit ourselves to those states which we expect to be converted into $\hat{S}_y$ and $2\hat{S}_x\hat{I}_z$ coherences, and thus yield oscillatory components in the real parts of the DEERATom traces detected at $\nu_S$ (Supplementary Table 4). Experimentally the filter pulses were applied at a time such that their centre corresponded to the maximum echo intensity, and we mimic this in our simulations. The fidelity of the operations is determined by comparing the quantum state at time $2\tau^+$ to the expected coherence. For the 'single molecule' model system, in which both spins are directly on resonance with the applied pulses, the simulated fidelity decreases uniformly as the pulse length is increased. For 1 ns pulses the fidelity of all pairs of operations is 1.0000, but this drops to between 0.9788 and 0.9882 for a pulse length of 80 ns (Fig. 8a, b). There is no discernible pattern between the pairs of operations and their sensitivity to pulse length.

Just as for the simulations of the entangling gate, the fidelity of the filter pulses is highly dependent on the spectral linewidth and drops to around 0.6 as the linewidth exceeds the pulse bandwidth (Fig. 8c, d). This decrease is approximately the same for both in- and anti-phase coherences and for all operator pairs. As the difference between the resonance frequency of a spin packet and the frequency of the microwave pulse increases, the spin is less affected by the pulse and

retains more of its original magnetisation. As such the application of filter pulses to broad spectra results in a statistical mixture of excited and unexcited magnetisation.

**Detection.** The influence of pulse length, coupling frequency distributions, and spectral linewidths on the traces recorded by pump-detect block are well studied in the context of dipolar spectroscopy[26,40]. While these affect the form factor of the time traces, we do not need to consider them here as all DEERATom traces were recorded with the same pulse parameters and so will be affected equally. Thus, for high-fidelity readout the pump-detect block must be equally sensitive to in- and anti-phase coherences. This is conveniently demonstrated by simulating traces for which the initial state is either a pure in- or anti-phase coherence on the detection spin, using the "single molecule" model from previous simulations (Fig. 9). From this we conclude that the pump-detect block is an effective way to measure the expectation values of both in- and anti-phase coherences.

**Relation to experiment.** The simulations presented thus far have considered the absolute quantum state of the system at the end of the relevant process normalised relative to the initial state. It is not possible to normalise the experimentally determined quantum state in this way as the initial state is not determined. Instead, the fitted state was normalised by the magnitude of the calculated expectation values, and so is insensitive to an overall loss in magnetisation. This relative normalisation process can dramatically increase the experimental

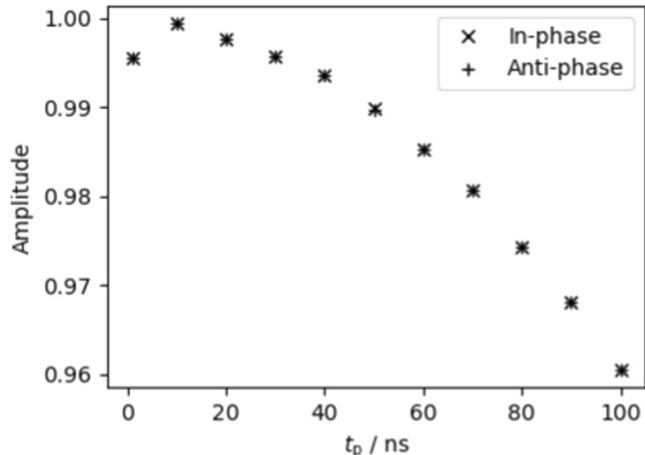

**Fig. 9 |** The amplitude of the oscillations in simulated time traces as a function of pulse length, within-phase coherences leading to cosine oscillations (crosses) and anti-phase coherences leading to sine oscillations (plusses).

fidelity, especially for the simulations of the entangling gate considering anisotropic coupling frequencies. The uniformly and orientationally weighted simulations with $\omega_\perp = 2\,\text{MHz}$ have simulated pseudo-fidelities of just 0.746 and 0.820, but normalised pseudo-fidelities of 0.963 and 0.927.

Similarly, the simulations considering distributions in resonance frequencies were calculated assuming that all pairs of resonance frequencies contribute equally to the quantum state. While this is obviously true in the sample as a whole, only a sub-population of the sample contributes to the DEERATom time traces, with the contribution of each spin pair weighted by the excitation fraction of the individual qubits by the detection and pump pulses. As such much of the loss in fidelity demonstrated by the entangling gate and filter pulses (Figs. 7 and 8) will not be detected by the pump-detect block.

**Decoherence.** To reduce the computational cost of these simulations they were performed in the Hilbert space, rather than the exponentially more expensive Liouville space in which relaxation models are more readily implemented. As such we note that the maximal reduction signal intensity would arise from the need to maintain coherence from the initial pulse up to the detection. Experimentally these two events are 3.5 μs apart, and thus we would expect a 30% loss of coherence across the entire experiment using the values from Supplementary Fig. 3. While this may impact the DFHE entangling gate, and thus the magnetisation present at $2\tau$, it does not significantly affect the DEERATom readout as all traces require coherences to be maintained for the same length of time. Significantly longer coherence times have been reported[13] in molecular electron spin qubits, and the extension and impact of coherence times, being primarily a single-qubit phenomenon, is beyond the scope of this work.

We have investigated the feasibility of transferring well established NMR quantum computing protocols to EPR, specifically regarding the implementation of multi-qubit gates required for quantum computing. To this end we implemented an entangling gate by pulse EPR, as well as proposing and implementing a novel density matrix tomography scheme tailored to the hierarchy of interaction strengths found in multi-qubit molecular electron spin systems. Experiments in the high temperature regime can only ever demonstrate pseudo-entanglement but working in this regime does allow us to test the protocols and determine the fidelities of its various component operations. For sufficiently high fidelities, this procedure could be used to demonstrate real entanglement at lower temperatures.

By simulating the entangling gate and readout separately we were able to assess the impact of key factors on the fidelity of both aspects

of the experiment separately. While long pulse lengths do have a detrimental effect on both state generation and detection, it is the distribution of coupling frequencies and partial excitation of near resonant spin which have the largest impact. These factors will not affect the manipulation of single molecules, but they are significantly deleterious to proof-of-concept ensemble experiments; the fidelities presented here are inadequate to demonstrate real entanglement, even at low temperatures. The distribution in coupling frequencies could be reduced by engineering molecules in which isotropic through-bond coupling is much larger than the anisotropic through-space interactions, however this will not be trivial as the through-bond interactions are generally negligible in the distance regime accessible by pulse dipolar spectroscopy ($\gtrsim 1.5\,\text{nm}$)[23,39]. As such we consider it more feasible to reduce the spread of couplings by orienting the ensemble, either in a single- or liquid-crystal.

Orienting the sample would also narrow the spectra of anisotropic spin systems as angular distributions of spin Hamiltonian parameters will also be reduced to a single value. The oriented spectrum will still be broadened by couplings to neighbouring electron and nuclear spins, as well as the presence of different molecular conformations which will introduce distributions in $g$-values, hyperfine couplings, and zero-field splitting parameters where relevant[41]. As such, it is likely that the reduction of partial excitation of near resonant spin packets will require both orientation of the sample and enhanced uniformity of excitation. The bandwidth of rectangular pulses can be increased by reducing pulse lengths, however sufficiently broadband pulses would need to be of picosecond length. Not only is this not attainable using available hardware, it is also undesirable as such short pulses would not be able to manipulate individual qubits independently. A preferable approach would be to use shaped pulses with more rectangular inversion profiles, either capable of exciting the entire EPR spectrum or with steep band-flanks to minimise partial excitation[40,42,43].

## Methods

All pulse EPR data were collected using a Bruker E580 spectrometer equipped with a SpinJet Arbitrary Waveform Generator and MD5 resonator at 50 K on a 0.2 mM solution of compound **1** made up in a solution of equal parts toluene and chloroform. Echo-detected field-swept spectra were collected using a Hahn-echo sequence $t_{90°} - \tau - t_{180°}$, with $t_{90°} = 20\,\text{ns}$, $t_{180°} = 40\,\text{ns}$, and $\tau = 400\,\text{ns}$. All frequency-static experiments were performed at 342.9 mT, except the 2P-ESEEM and amplifier characterisation which were performed at 342.7 mT. Transient nutations for resonator and amplifier characterisation used the pulse sequence $t - T - t_{90°} - \tau - t_{180°}$ with $t_{90°} = 20\,\text{ns}$, $t_{180°} = 40\,\text{ns}$, $\tau = 800\,\text{ns}$, $T = 25\,\mu\text{s}$, and an initial $t = 0\,\text{ns}$. 2P-ESEEM data were collected using the sequence $t_{90°} - \tau - t_{180°}$ with $t_{90°} = 100\,\text{ns}$, $t_{180°} = 200\,\text{ns}$, and an initial $\tau$ of 1 μs. DFHE-decay data were collected using the sequence $t_{90°} - \tau - t_{180°}$, $t_{90°} = 80\,\text{ns}$, $t_{180°} = 80\,\text{ns}$ and an initial $\tau$ of 0 ns. 3P-DEER data used the detection sequence $t_{90°} - \tau - t_{180°}$, with $t_{90°} = 80\,\text{ns}$, $t_{180°} = 80\,\text{ns}$ and $\tau = 1.5\,\mu\text{s}$, using an 80 ns pump pulse. 5P-DFHE-DEERATom data used the detection sequence $t_{90°} - \tau - t_{180°} - \tau - t_{var} - T - t_{180°} - T$ with $t_{90°} = 80\,\text{ns}$, $t_{180°} = 80\,\text{ns}$, $\tau = 250\,\text{ns}$, and $T = 1.5\,\mu\text{s}$ and with an 80 ns pump pulse, and $t_{var} = 80\,\text{ns}$ when active. All delay times are defined to be between the start of consecutive pulses.

## Data availability

Source data for this paper are available at https://doi.org/10.48420/24158442.

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

## Acknowledgements

This work was supported by the EPSRC (EP/P000444/1 and EP/R043701/1/) including an Established Career Fellowship (EP/R011079/1) to REPW. REPW also thanks the European Research Council for an Advanced Grant (ERC-2017-ADG-786734). We also thank the EPSRC for access to the EPR National Research Facility (EP/W014521/1, NS/A000055/1, EP/V035231/1 and EP/S033181/1). This work has received funding from the European Union's Horizon 2020 Program under Grant Agreement No. 862893 (FET-OPEN project FATMOLS). A.M.B. is grateful to The Royal Society and EPSRC for a Dorothy Hodgkin Fellowship (DH160004), and the University of Manchester for a Dame Kathleen Ollerenshaw Fellowship. A.M.B. also thanks the Royal Society of Chemistry for a Community for Analytical and Measurement Science fellowship (CAMS Fellowship 2020 ACTF ref. 600310/09). A.M.B. and C.J.R. thank The Royal Society for the Enhancement Award (RGF\EA\180287) which funded the doctoral stu-dentship for C.J.R.

## Author contributions

E. J. L. measured the EPR spectra discussed, advised by A. M. B. and E. J. L. M. Synthesis was by C. J. R. Data analysis was by E. J. L. and J. M.,

advised by J. L., A. A. and A. M. B. The project was devised by A. A., A. M. B. and R. E. P. W. The manuscript was written by E. J. L. and R. E. P. W., with input from all authors.

## Competing interests

The authors declare no competing interests.
