## [Peer Review File · Nature Communications]

REVIEWER COMMENTS

Reviewer #1 (Remarks to the Author):

The manuscript from Edmund J. Little et al. investigates the experimental realization of entangling two qubits in molecular electron spin qubits embodied in bis-nitroxide by means of multifrequency Electron Paramagnetic Resonance (EPR). Experimentally, a series of pulse sequences are applied for entanglement generation both in liquid and frozen solutions. The main finding in their manuscript is that they have successfully transferred entanglement generation through NMR to multifrequency EPR spectroscopy. The readout of the entangled state is performed using quantum state tomography (QST). The authors found a significant lack of fidelity in the entangled states, which they reasoned was due to the presence of isotropic exchange couplings and anisotropic dipolar couplings.

Physical implementation of quantum gates is quite a challenging field; this requires an advanced understanding of spectroscopy and quantum information processing. Overall, the present work is solid, and the manuscript is well written. Also, the theoretical investigation (especially QST simulations) provides useful insight to increase the fidelity of quantum gates in future experiments.

Thus, I support the publication of the present manuscript in nature communication after the author addresses the following comments:

- Is this experimental implementation scheme limited to only entangling two qubits? The dipolar qubit-qubit couplings are permanent and cannot be turned off, which makes scalability a huge problem in this scheme. Can the author provide a strategy for scalability for entangling more than two qubits?
- In figure 8a, the largest components of QST are $2S_{xlx}$ and $2S_{z lz}$. Do these components remain dominant with respect to different sample orientations? The inter-spin coupling will be modified with respect to sample orientation upon applying a magnetic field. In an optimal orientation where inter-spin couplings are such that the $2S_{xlx}$ and $2S_{z lz}$ are minimized, the fidelity will be increased.
- In the initialization step, the initial state is not determined in the experiment. Can the author describe the nature of the starting state in terms of the spin basis? Also, authors should provide the evolution of eigenstates of the spin Hamiltonian vs. the magnetic field.

- Molecules come with acoustic and optical phonon degrees of freedom, which can induce decoherences even at low temperatures. Is it possible that the qubit coherences are already destroyed during the application of EPR pulses? Hence the fidelity of the entangled state?

- Authors should cite: Phys. Rev. Lett. 93, 206603, where pseudo-entanglements of spin states were generated, which is in fact an extension of the current citation 8.

Discussion and conclusion are separate for some reason; they should be combined.

Reviewer #2 (Remarks to the Author):

Authors have characterized entangling gate on a single-molecule spin system, with qubits encoded into electron and nuclear spin. In a series of electron spin resonance experiments, they have assessed interactions that produce the gates and impediments to the gate implementation that come both from the studied molecule and the experimental setup. Results show that the interactions needed for gate implementation are indeed present in the molecule and that manipulation of their effects by applying resonant pulses behave as expected. In addition to these encouraging findings, they find and describe how imperfect pulses, insufficient precision in state preparation and detection, and uncontrollable variations in the system properties over the ensemble of measured molecules affect the gate implementations.

I find the results presented in the manuscript very interesting and useful. The art of quantum state tomography with pulse design, as well as the description of what is achievable and what is not are quite illuminating. Insights from the experimental experience into causes and consequences of limitations of methods and equipment and modeling of resulting signals appear correct to the best of my knowledge. The description of models, system, procedures, data handling and theoretical modeling are of the adequate level, making the manuscript readable for a nonspecialist. Therefore, I would recommend the manuscript for publication, after the following comments have been addressed.

Introduction, pg. 2

The sentence: 'It must also be possible to detect the quantum state of the qubits, which can be accomplished by quantum state tomography, in which the expectation values of a set of operators which form a complete basis of the Hilbert space are measured.' is correct, but may appear confusing on first reading. Wording suggests that quantum state tomography is needed for quantum computing. This is not true, and in various schemes requirements are different. The most standard requirement for measurements is to be able to measure in logical basis. This is less than tomography. More important point is that ESR with appropriate pulse design is sufficient for tomography. In my opinion, precise statement of this kind would make the manuscript more readable.

Introduction, pg. 3

Authors emphasize reproducible spin interactions and bottom-up design as advantages, of molecules as spin qubits. This is correct, but obscures a point that a quantum computing needs to be implemented on a molecule with many spins that can be individually addressed by the pulses. This implies that molecules carrying more and more qubit get more and more complex. The problem of scaling should also be mentioned in this context.

Background, pg. 5

First sentence says that maximally entangled states are Bell states. This is not clearly correct. True description is that Bell basis consists of maximally entangled states.

Background, pg. 6

Intuition about resonance methods that detect pseudo entanglement would be very helpful. A reader will be thankful. The point to be passed is that resonant techniques allow detection of small deviations from infinite temperature density matrix.

Background, pg. 8

The phrase 'FID signal', with citation to [23] appears for the first time as an abbreviation.

Background, pg. 9

Table caption and text: The authors talk about tomography which requires measurement of 15 parameters for two-qubit density matrix, as correctly stated. It is not clearly elaborated what constitutes 'spin operators which lead to coherences', neither in the caption nor in the corresponding portion of the main text.

Background, pg. 9

Equation 2 is referred to, but equations are not labeled in the manuscript.

Background, pg. 10

Discussion of pump refocus explains the table and relations between measured coherences and tomography. It is not explicit enough, counting of operators should be in the text.

DEER assisted tomography, pg. 18

Authors reference 'section 9.1'. What does it mean?

DEER assisted tomography, pg. 22

Authors reference 'section 4'. What does it mean?

DEER assisted tomography, pg. 22

typo: (Figures 8c and) Should it be 8c and 8d ?

Supplementary material, pg. 8

Phrase 'Error! Reference source not found' appears in the manuscript.

Supplementary, pg. 17

Same as above

Response to Reviewer Comments

We thank the reviewers for their thorough reviews and for their helpful suggestions. We have accepted all their comments. Answers to specific points below. Apologies that it is quite lengthy.

Reviewer #1:

The manuscript from Edmund J. Little et al. investigates the experimental realization of entangling two qubits in molecular electron spin qubits embodied in bis-nitroxide by means of multifrequency Electron Paramagnetic Resonance (EPR). Experimentally, a series of pulse sequences are applied for entanglement generation both in liquid and frozen solutions. The main finding in their manuscript is that they have successfully transferred entanglement generation through NMR to multifrequency EPR spectroscopy. The readout of the entangled state is performed using quantum state tomography (QST). The authors found a significant lack of fidelity in the entangled states, which they reasoned was due to the presence of isotropic exchange couplings and anisotropic dipolar couplings.

Physical implementation of quantum gates is quite a challenging field; this requires an advanced understanding of spectroscopy and quantum information processing. Overall, the present work is solid, and the manuscript is well written. Also, the theoretical investigation (especially QST simulations) provides useful insight to increase the fidelity of quantum gates in future experiments.

Thus, I support the publication of the present manuscript in nature communication after the author addresses the following comments:

- Is this experimental implementation scheme limited to only entangling two qubits? The dipolar qubit-qubit couplings are permanent and cannot be turned off, which makes scalability a huge problem in this scheme. Can the author provide a strategy for scalability for entangling more than two qubits?

Answer:

The scheme described in this manuscript does indeed exploit an always-on interaction that is weak compared to other energy scales in the system to controllably entangle the two intramolecular electron spin qubits and verify the entanglement. This is an objective specific to the family of two-qubit molecules.

The reviewer is right that scaling up is likely to require additional ingredients; a particularly important one is a controllable inter-qubit interaction. A range of options is being explored by the authors and the wider community.

One approach is to develop the theme of “engineerable” molecular components to include optically switchable units in the inter-qubit coupling. (The group of the corresponding author is currently investigating optically switchable bridging units between Cr7Ni antiferromagnetic

molecular spin qubits.) Another approach is to strongly couple a collection of frequency-tuneable molecular spin qubits to a resonator mode. (We have previously shown that, with appropriate design, individual molecular spin qubits can be tuned using local electric fields.) By tuning pairs of qubits into resonance with the resonator, an entangling interaction between them is mediated by virtual photon exchange.

These are just two examples of the many directions being explored by the wider community for integration and scaling of molecular-spin-based quantum technologies. All such efforts will be boosted by the demonstration of pairwise entanglement of the kind addressed by our manuscript.

- In figure 8a, the largest components of QST are $2S_x I_x$ and $2S_z I_z$. Do these components remain dominant with respect to different sample orientations? The inter-spin coupling will be modified with respect to sample orientation upon applying a magnetic field. In an optimal orientation where inter-spin couplings are such that the $2S_x I_x$ and $2S_z I_z$ are minimized, the fidelity will be increased.

Answer

The result presented in figure 8a is for experimental results measured on a frozen solution. We did not have a signal crystal or orientated sample available to us to conduct this work, although this would be an interesting proposal for future work if it is possible chemically to generate such a sample. Therefore, it is not possible to test experimentally at this time if the $2S_x I_x$ and $2S_z I_z$ components remain dominant in an orientated sample. The reviewer is correct that in an orientated sample the inter-spin coupling will be modified but the degree to which it is modified will depend on the degree of orientation ordering that can be achieved. It is expected that in an ideal ordered system it may be possible to find an optimal orientation where the inter-spin couplings are such that the $2S_x I_x$ and $2S_z I_z$ components are minimised and that this would increase fidelity. However, at this time it has not been possible to generate an ordered system using the molecule used in this study and thus such experiments fall beyond the scope of this study.

- In the initialization step, the initial state is not determined in the experiment. Can the author describe the nature of the starting state in terms of the spin basis? Also, authors should provide the evolution of eigenstates of the spin Hamiltonian vs. the magnetic field.

Answer

The experimental starting state is determined by the Boltzmann distribution. This yields a state with an excess magnetization along the z axis. As stated in the manuscript the density matrix under thermal equilibrium is:

$$\hat{\rho}_{eq} = (1/2) \mathbb{1} + \Delta p [S_z + I_z] + \Delta p^2 2S_z I_z$$

this is the starting state for the experiment.

The reviewer asks for the evolution of eigenstates of the spin Hamiltonian vs the magnetic field. This request is a little unclear - we presume that this refers to the plots shown in figure 12 which show how the fidelity of the operator pairs vary with pulse length and spectral linewidth?

Experimentally as the spectrum of the system studied is a limited width, changing the field alone will cause the pulses used to shift from resonance with this spectrum. As a result, both a change in frequency and field is required to remain on resonance. Experimentally this change in field and frequency causes a change in the experimental spectral shape due to g-tensor anisotropy and consequently keeping a fixed offset between the pulses may cause different orientations of each centre to be on resonance with the pulses, which may also have impact on the fidelity.

In our experiment and simulations this orientation selection was constant as a constant field and frequency used. Consequently, a simulation in which field is changed may not be representative of the experimental system unless orientation selection is considered, and at the same time considering orientation selection means that the changes seen are not only a result of changing the field. For this reason, as the orientation selection is a function of the field, the molecular ordering and molecular flexibility and is therefore complex and it is not possible to capture in a single calculation. It is expected that increasing the field will increase the fidelity.

- Molecules come with acoustic and optical phonon degrees of freedom, which can induce decoherences even at low temperatures. Is it possible that the qubit coherences are already destroyed during the application of EPR pulses? Hence the fidelity of the entangled state?

Answer

Traditional time-resolved magnetic resonance experiments on ensembles of individual molecular spin qubits (diluted so as to remove the through-space inter-qubit interactions) provide rich data on the various factors contributing to energy relaxation and phase decoherence. Vibrational degrees of freedom are indeed among a selection of important mechanisms at play in molecular spin qubits.

Energy relaxation (on the timescale commonly referred to as T_1 , and sometimes described as “spin-lattice relaxation”) is often dominated by vibrational couplings. In general it is found that at low temperatures T_1 increases very rapidly as the population of thermal vibrations declines, confirming that the spin-phonon coupling is too small for spontaneous phonon emission to be an important process.

Phase coherence (with the timescale T_m) also tends to increase at low temperatures, and, in this limit, is often dominated by environmental nuclear spin dynamics.

Magnetic molecules qualify for description as “molecular spin qubits” based in part on their long single-qubit relaxation and decoherence times. In our system, T_1 is very long and T_m is about 7 microseconds (see SI section 3.2). Our protocol is designed around the hierarchy of energy scales usually encountered in such two-molecular-spin-qubit molecules: $h/T_m > J > h/T_{(manip.)}$ where T_m is the single-qubit phase coherence time, J is the inter-qubit interaction energy, and $T_{(manip.)}$ is

the single-qubit operation time. We have added a sentence in the introduction to clarify this point.

In the system we report, T_m is about 7 microseconds, exceeding the duration of the experiment. We can be confident, therefore, that that the entanglement fidelity loss is driven by other mechanisms, principally, we believe, variations in J across the measured ensemble.

- Authors should cite: Phys. Rev. Lett. 93, 206603, where pseudo-entanglements of spin states were generated, which is in fact an extension of the current citation 8.

Answer

We have added this reference and renumbered the following references.

Discussion and conclusion are separate for some reason; they should be combined.

Answer

As the discussion is lengthy we feel it is better to recap the discussion in a conclusion section. We have not followed this advice.

Reviewer #2 (Remarks to the Author):

Authors have characterized entangling gate on a single-molecule spin system, with qubits encoded into electron and nuclear spin. In a series of electron spin resonance experiments, they have assessed interactions that produce the gates and impediments to the gate implementation that come both from the studied molecule and the experimental setup. Results show that the interactions needed for gate implementation are indeed present in the molecule and that manipulation of their effects by applying resonant pulses behave as expected. In addition to these encouraging findings, they find and describe how imperfect pulses, insufficient precision in state preparation and detection, and uncontrollable variations in the system properties over the ensemble of measured molecules affect the gate implementations.

I find the results presented in the manuscript very interesting and useful. The art of quantum state tomography with pulse design, as well as the description of what is achievable and what is not are quite illuminating. Insights from the experimental experience into causes and consequences of limitations of methods and equipment and modeling of resulting signals appear correct to the best of my knowledge. The description of models, system, procedures, data handling and theoretical modeling are of the adequate level, making the manuscript readable for a nonspecialist. Therefore, I would recommend the manuscript for publication, after the following comments have been addressed.

Introduction, pg. 2

The sentence: 'It must also be possible to detect the quantum state of the qubits, which can be accomplished by quantum state tomography, in which the expectation values of a set of operators which form a complete basis of the Hilbert space are measured.' is correct, but may appear confusing on first reading. Wording suggests that quantum state tomography is needed for quantum computing. This is not true, and in various schemes requirements are different. The most standard requirement for measurements is to be able to measure in logical basis. This is less than tomography. More important point is that ESR with appropriate pulse design is sufficient for tomography. In my opinion, precise statement of this kind would make the manuscript more readable.

Answer

We have rephrased this sentence to try and make it clearer.

Introduction, pg. 3

Authors emphasize reproducible spin interactions and bottom-up design as advantages, of molecules as spin qubits. This is correct, but obscures a point that a quantum computing needs to be implemented on a molecule with many spins that can be individually addressed by the pulses. This implies that molecules carrying more and more qubit get more and more complex. The problem of scaling should also be mentioned in this context.

Answer

We have added a sentence to the text to draw attention to scaling

Background, pg. 5

First sentence says that maximally entangled states are Bell states. This is not clearly correct. True description is that Bell basis consists of maximally entangled states.

Answer

This has been corrected to say this includes the Bell states....

Background, pg. 6

Intuition about resonance methods that detect pseudo entanglement would be very helpful. A reader will be thankful. The point to be passed is that resonant techniques allow detection of small deviations from infinite temperature density matrix.

Answer

The sentence 'Resonance techniques are able to detect pseudo-entanglement due to their sensitivity to small deviations from the equilibrium density matrix.' Has been added to this section.

Background, pg. 8

The phrase 'FID signal', with citation to [23] appears for the first time as an abbreviation.

Answer

The acronym has been expanded.

Background, pg. 9

Table caption and text: The authors talk about tomography which requires measurement of 15 parameters for two-qubit density matrix, as correctly stated. It is not clearly elaborated what constitutes 'spin operators which lead to coherences', neither in the caption nor in the corresponding portion of the main text.

Answer

The table caption has been changed to read: Table 1: Spin operators which lead to single quantum coherences, detectable as transverse magnetization (top) under all pairs of operations. The subscripts on the operations denote the spins on which the operation is applied. We now define these coherences as single quantum coherences that are detectable as transverse magnetization.

Background, pg. 9

Equation 2 is referred to, but equations are not labeled in the manuscript.

Answer

*The equations have now been numbered. The equation in question is now **equation X***

Background, pg. 10

Discussion of pump refocus explains the table and relations between measured coherences and tomography. It is not explicit enough, counting of operators should be in the text.

Answer

The table caption has been changed as detailed above, and the text in the manuscript modified to read: 'The presence of in-phase coherences, those which would be detectable as transverse

magnetization, before the 'pump-refocus' block will manifest as cosine oscillations in the intensity of the refocussed echo and anti-phase coherences will manifest as sine oscillations, with the oscillation amplitudes proportional to their expectation values (see the product operator calculations in SI 11 and their relation to expectation values in SI 9.1).'

DEER assisted tomography, pg. 18

Authors reference 'section 9.1'. What does it mean?

Answer

This refers to Section S2.1 in the supplementary information. It has been corrected

DEER assisted tomography, pg. 22

Authors reference 'section 4'. What does it mean?

Answer

This refers to the Discussion section. It has been corrected.

DEER assisted tomography, pg. 22

typo: (Figures 8c and) Should it be 8c and 8d ?

Answer

Yes. This has been corrected.

Supplementary material, pg. 8

Phrase 'Error! Reference source not found' appears in the manuscript.

Answer

This has been corrected.

Supplementary, pg. 17

Same as above

Answer

This has been corrected.

REVIEWERS' COMMENTS

Reviewer #1 (Remarks to the Author):

I thank the authors for their kind reply. The specific points that I had raised are now clearer. I thus recommend its publication in nature communication.

Reviewer #2 (Remarks to the Author):

Authors have satisfactory answered my questions and changed the manuscript accordingly. In my opinion, the present version of the manuscript should be published.